# Simultaneous Visualization of MiRNA-221 and Caspase-3 in Cancer Cells for Investigating the Feasibility of MiRNA-Targeted Therapy with a Dual-Color Fluorescent Nanosensor

**DOI:** 10.3390/bios12070444

**Published:** 2022-06-23

**Authors:** Mingyao Ren, Zhe Chen, Chuandong Ge, Wei Hu, Nianxing Wang, Limin Yang, Mingming Luan, Jing Xu

**Affiliations:** 1Shandong Provincial Key Laboratory of Molecular Engineering, School of Chemistry and Chemical Engineering, Qilu University of Technology (Shandong Academy of Sciences), Jinan 250353, China; rmyao1997@163.com (M.R.); 17861405811@163.com (Z.C.); gechuandong2021@163.com (C.G.); weihu@qlu.edu.cn (W.H.); wnx@qlu.edu.cn (N.W.); 2College of Chemistry and Pharmaceutical Sciences, Qingdao Agricultural University, Qingdao 266109, China; yanglimin@qau.edu.cn

**Keywords:** fluorescent nanosensor, miRNA-221, apoptosis, caspase-3, cell imaging

## Abstract

MiRNA-targeted therapy holds great promise for precision cancer therapy. It is important to investigate the effect of changes in miRNA expression on apoptosis in order to evaluate miRNA-targeted therapy and achieve personalized therapy. In this study, we designed a dual-color fluorescent nanosensor consisting of grapheme oxide modified with a molecular beacon and peptide. The nanosensor can simultaneously detect and image miRNA-221 and apoptotic protein caspase-3 in living cells. Intracellular experiments showed that the nanosensor could be successfully applied for in situ monitoring of the effect of miRNA-221 expression changes on apoptosis by dual-color imaging. The current strategy could provide new avenues for investigating the feasibility of miRNA-targeted therapy, screening new anti-cancer drugs targeting miRNA and developing personalized treatment plans.

## 1. Introduction

Cancer is a ubiquitous and devastating disease, which has led to many cancer diagnosis- and treatment-inspired research fronts focused on the creation of various detection and therapy platforms. Precision therapy provides remarkable advantages in cancer treatment due to its high cure rate, low side effects and high precision [1]. Precision therapy can uncover pathogenesis and offer personalized treatment according to the individual’s genetic information, lifestyle and living environments [2]. The importance of personalized treatment has risen to the national strategic level, and miRNA-targeted therapy is an active research field in precision medicine.

MiRNAs are small and non-coding RNAs that play critical regulatory roles in early development, physiology and pathological conditions. Aberrant expression levels of miRNAs are associated with the initiation and development of various cancers [3]. MiRNAs may play different roles as oncogenes or tumor suppressors in the human body [4,5,6,7]. Studies have shown that apoptotic genes could be regulated to induce cell apoptosis by enhancing or inhibiting the expression of specific miRNA [8,9]. In this regard, miRNAs are candidates for precision cancer therapy. Thereby, it is necessary to understand the effect of changes in miRNAs’ expression on apoptosis when studying miRNA-targeted precision cancer therapy. 

Traditional approaches for detecting miRNAs (such as Northern blotting, microarray analysis and qRT-PCR) and apoptosis (such as electron microscopy, immunohistochemical analysis and Western blotting) all require cell fixation and lysis [10,11,12,13,14]. Hence, these approaches cannot really reveal the expression changes of miRNA and apoptotic markers in living cells. Moreover, these methods also have the disadvantage of being tedious and time-consuming experimental operations. Recently, fluorescent nanoprobes have become a research focus in many fields [15,16,17,18], and they have been widely used for the detection of miRNAs and apoptotic markers in living cells due to having a large number of advantages, including being non-invasive, visual identification, and extremely small consumption [19,20,21,22,23,24,25,26,27]. However, most fluorescent imaging approaches can achieve single detection of intracellular miRNA or apoptotic markers so that they cannot visually detect the effect of miRNA expression changes on cell apoptosis. Therefore, simultaneous detection of miRNA and apoptosis is of great significance in order to visualize miRNA regulation of apoptosis. 

At present, many nanomaterials have been widely used for biosensing and imaging because of their good biocompatibility and strong fluorescence quenching ability, including gold nanoparticles [28,29], polydopamine [30,31], molybdenum disulfide [32,33] and grapheme oxide (GO) [34,35]. Among them, GO is a two-dimensional structure consisting of hexagonal arrangements of sp^2^-hybridized carbon atoms [36], with a large specific surface area and better dispersion in a wide range of solvents [37]. This makes it an ideal carrier for adsorbing biomolecules. It has been reported that GO can be functionalized with a wide range of biomolecules such as DNA, peptide chains and proteins by physical adsorption or chemical conjugation [38]. In addition, it has been pointed out that the planar carbon π system of GO can efficiently quench all types of fluorescent dyes through the photoinduced electron transfer process [39]. Hence, GO can be used as an excellent fluorescence quenching agent for fluorescence biosensors in vivo and in vitro. Herein, we reported a dual-color fluorescent nanosensor for the simultaneous detection and imaging of miRNA-221 and apoptotic marker caspase-3 in living cells (Figure 1). In this study, the fluorescent nanosensor was developed by gradually modifying GO with a molecular beacon (MB) and a peptide. GO was selected as a good carrier to absorb the MB and peptide via the π–π stacking effect [40]. The MB had a specific stem-loop structure labeled with Cy5 and could specifically recognize the target miRNA-221. The peptide was targeted to caspase-3, which was labeled with FITC. Generally, in the absence of the targets, the fluorescent signals of the two dye molecules attached with MB and peptide were well quenched by GO. Upon meeting caspase-3 and miRNA-221, the peptide was cleaved and the MB was forced to open, thereby leading to the restoration of the fluorescence.

## 2. Materials and Methods

### 2.1. Fluorescence Quenching

To optimize the amount of GO, Cy5-modified MB (100 nM) was added to different concentrations of GO (PBS, pH 7.4). The fluorescent signal of Cy5 was recorded with λex/λem = 648/667 nm. FITC-modified peptide (300 nM) was added to various concentrations of GO-MB solutions (PBS, pH 7.4). The fluorescent signal of FITC was collected with λex/λem = 480/520 nm to determine the optimal amount of GO-MB. 

### 2.2. Preparation of the Fluorescent Nanosensor

In total, 100 nM MB solution (PBS, pH 7.4) was added to 80 μg/mL GO solution and shaken for 1 h at room temperature. Then 300 nM peptide solutions (PBS, pH 7.4) was mixed with the above solution and further incubated for 1 h. The following mixture was further purified through repeated centrifugation and the precipitate was dispersed in PBS to obtain the fluorescent nanosensor.

### 2.3. Fluorescence Response of the Nanosensor

For the detection of miRNA-221, the nanosensor (80 μg/mL) was incubated with different concentrations of the perfectly matched miRNA-221 targets (0, 50, 100, 150, 200, 300, 400, 500, 1000, 2000, 2500 nM) for 1 h at 37 °C. The fluorescence of Cy5 dye was collected with 648 nm excitation and 667 nm emission. For the detection of caspase-3, the fluorescent nanosensor (80 μg/mL) was added to various concentrations of recombinant caspase-3 (0, 0.5, 1, 2, 3 and 4 units). After incubation for 4 h at 37 °C, the fluorescence of FITC dye was observed with 480 nm excitation and 520 nm emission. 

### 2.4. Kinetic Experiments

To optimize the response time, the fully matched miRNA-221 targets (2000 nM) were added into the nanosensor solution (80 μg/mL). The fluorescence was collected with increasing incubation time (0, 10, 20, 30, 40, 50, and 60 min) at 37 °C with 648 nm excitation and 667 nm emission. The recombinant caspase-3 (4 units) was mixed with the nanosensor solution (80 μg/mL) and incubated for various times (0, 0.5, 1, 2, 3, and 4 h) at 37 °C. The fluorescence intensity of FITC-labeled peptide was collected at each time point with 480 nm excitation and 520 nm emission. 

### 2.5. Specificity of the Nanosensor

To detect the targeting specificity of the nanosensor toward miRNA-221, the perfectly matched miRNA-221 target and other targets were investigated. The concentration of each target is 1000 nM. After culturing with the nanosensor (80 μg/mL) for 1 h at 37 °C, the fluorescence intensity of Cy5 was collected. To study the selectivity for caspase-3 detection, caspase-3 (4 units) and other interferents including 4 μg/mL hemoglobin (HGB), 4 μg/mL bovine serum albumin (BSA), 20 mM glucose (Glu), 2 mM CaCl_2_ and 100 mM NaCl were added to the nanosensor (80 μg/mL). The fluorescence intensity of FITC was recorded after incubation for 4 h at 37 °C. 

### 2.6. Confocal Fluorescence Imaging

For the confocal imaging of caspase-3 and miRNA-221 in living cells with drug treatment, A549 and Hela cells were seeded in three confocal dishes and incubated for 24 h, respectively. Then one group of cells was treated with 10 μg/mL lipopolysaccharide (LPS) and another group of cells were treated with 0.1 μM staurosporine (STS) for 12 h, respectively. The cells without treatment served as the control. After the nanosensor (80 μg/mL) was added to the three groups of cells for 4 h, the confocal laser scanning microscopy (CLSM) was then captured. Green channels (caspase-3) and red channels (miRNA-221) were excited at 488 and 633 nm, and the fluorescence was monitored between 500–600 nm and 660–700 nm, respectively.

To investigate the effect of miRNA-221 level on apoptosis in living cells with transfection reagents treatment, A549 and Hela cells were sorted into three confocal dishes to obtain a cell density of 30–50% at the time of transfection, respectively. According to manufacturer’s instructions, one group was pretreated with 100 pmol miRNA-221 mimics and another group was pretreated with 500 pmol miRNA-221 inhibitors for 48 h, respectively. The group without transfection treatment was served as the control. After the 80 μg/mL nanosensor was added to the three groups of cells for 4 h, the CLSM imaging was performed using the 488 nm laser and 633 nm laser.

## 3. Results and Discussion

### 3.1. Preparation and Characterization of the Nanosensor

We first analyzed GO by Raman spectrometer. As shown in Appendix A, the D peak represents the defect of the carbon atomic crystal, indicating the formation of sp^3^ carbon in GO, and the G peak was assigned to the vibration of sp^2^-bonded carbon atoms. Then, the surface elemental composition of GO was analysed by X-ray photoelectron spectroscopy (XPS). As shown in Appendix A, it is mainly composed of carbon and oxygen. The oxygen-containing functional groups are mainly C-O-C, COOH and C=O. The GO and nanosensor were further analyzed by transmission electron microscopy (TEM). The result showed that the GO and nanosensor had a flaky structure with good dispersion and the size was less than 400 nm (Figure 1A). Furthermore, the atomic force microscopy (AFM) results showed that the size distribution of the GO and nanosensor was from 50 to 300 nm in width, which was consistent with the TEM data (Figure 2A,C). However, the thickness of the GO increased from 2 nm to 4 nm after MB and the peptides were modified on the surface of the GO (Figure 2B,D). In addition, Zeta potential analysis was examined to verify that the potentials of GO and the nanosensor were −6.42 mV and −0.668 mV, respectively (Appendix A). The XRD spectra measured in a range of 2θ from 5° to 70° (Appendix A) showed a diffraction peak at 2θ = 10.98° (GO) and 2θ = 10.14° (nanosensor). The diffraction peak shifted slightly toward a small angle and the peak intensity decreased after the GO was modified with the MB and peptide. All these results confirmed that the nanosensor was successfully prepared.

In order to realize the best recognizing property, the concentration of GO was further optimized. The MB (100 nM) solution was added to different concentrations of the GO solution, and then the perfectly quenched GO-MB was successful prepared. As shown in Figure 3A,C, the fluorescence of Cy5 labeled on the MB was completely quenched when the solution of GO was 50 μg/mL. Then, 300 nM of the peptide solution was added to different concentrations of GO-MB with the quenching of Cy5 fluorescence. As shown in Figure 3B,D, 80 μg/mL of the GO-MB solution perfectly quenched the fluorescence of FITC dye labeled on peptide. Hence, 80 μg/mL of GO was chosen for the following experiments. In order to study the stability of the nanosensor, the pH tolerance assay was further tested. As shown in Appendix A, the color of the nanosensor solution became darker when the treated pH value was above 8.0. What is more, the fluorescence of Cy5 labeled on the MB and FITC labeled on the peptide was enhanced significantly under alkaline conditions (Appendix A). The results indicated that the nanosensor was unstable under alkaline conditions, and the MB and peptide could detach from the GO. Therefore, considering the stability and biocompatibility of the nanosensor are prerequisites for its biological application, the nanosensor was prepared at pH 7.4.

### 3.2. In Vitro Studies of the Nanosensor

To investigate the capability of the nanosensor for simultaneously detecting the miRNA-221 and caspase-3, we next studied the response of the peptide to caspase-3 and the response of the MB to the perfectly complementary DNA target. As shown in Figure 4A,C, the fluorescence of the Cy5-labeled MB gradually increased with increases in the fully complementary target-221 concentrations (0–2500 nM). Similarly, Figure 4B shows that the fluorescent signal of the FITC-labeled peptide gradually increased with increasing concentrations of caspase-3 (0–4 unit). The fluorescent signal had linear correlations with the concentration of the caspase-3 target (Figure 4D). These results suggest that the response of the nanosensor toward the corresponding targets led to the fluorescent signal’s recovery. The kinetic results suggest that the fluorescent nanosensor responded rapidly to the fully matched DNA target within 10 min and reached equilibrium in 30 min (Figure 5A and Appendix A). The fluorescent response of the nanosensor to caspase-3 was also gradually increased with time and reached equilibrium after 3 h (Figure 5B and Appendix A). 

To evaluate whether the nanosensor could specifically respond to caspase-3 and miRNA-221, we analyzed the effects of other interferents on the nanosensor. As shown in Figure 5C, the mismatched target-221 and other targets all showed slightly enhanced fluorescence compared to the control. However, the Cy5-labeled MB presented a strong fluorescent response when the nanosensor met fully complementary target-221. Similar results were also observed when the fluorescent nanosensor encountered caspase-3 and other interferents (Figure 5D). These results demonstrate that the nanosensor could simultaneously and specifically detect caspase-3 and miRNA-221.

### 3.3. MTT Experiment

Hela and A549 cells were chosen to investigate the cytotoxicity of the nanosensor. As shown in Figure 6, the cell viabilities of both cell lines remained over 95% after culturing the cells with the nanosensor for 24 h, which suggested that the nanosensor had good biocompatibility and little cytotoxicity. The results demonstrate that the nanosensor could be used for intracellular imaging and detection.

### 3.4. Imaging of MiRNA-221 and Caspase-3 in Living Cells

Hela and A549 cells were chosen to investigate the feasibility of the fluorescent nanosensor simultaneously detecting the expression levels of intracellular caspase-3 and miRNA-221. It has been reported that miRNA-221 expression was especially high as a cancer marker in cancer cells, and caspase-3 expression was increased during cell apoptosis [41,42,43,44,45]. The two cell types were all pretreated with STS and LPS. STS can promote cell apoptosis and increase the expression of caspase-3 [46]. LPS can induce cells to generate a pro-inflammatory response and upregulate oncogenic miRNA-221 expression [47]. Hela cells were separated into three parallel groups. One group was pretreated with LPS and another group was pretreated with STS. The group without drug treatment served as a control. Three groups of cells were examined by CLSM with suitable laser transmitters after incubating with the fluorescent nanosensor (80 μg/mL) for 4 h. As shown in Figure 7A, compared to the untreated Hela cells, the red fluorescence signal for miRNA-221 was significantly increased and the green fluorescence signal for caspase-3 had no change in Hela cells with LPS treatment. However, the green fluorescent intensity was significantly enhanced and the red fluorescent intensity decreased in cells with STS treatment.

WB and RT-PCR experiments were also used to examine the expressions of caspase-3 and miRNA-221 in cells. Figure 7B,C show that the results of WB and RT-PCR maintained consistency with the results of CLSM. In addition, the CLSM, WB and RT-PCR methods were also examined in A549 cells and all the experimental results were similar to those detected in Hela cells (Figure 7D–F). The results show that the fluorescent nanosensor could simultaneously monitor intracellular miRNA-221 and caspase-3.

To evaluate the feasibility of the nanosensor monitoring the intracellular miRNA-221 regulation of apoptosis, Hela and A549 cells transfected with miRNA-221 mimics or inhibitors were then examined to investigate the effect of miRNA-221 expression on caspase-3. Hela cells were divided into three parallel groups. One group was pretreated with miRNA-221 mimics to increase the expression of miRNA-221 and another group was pretreated with miRNA-221 inhibitors to decrease the expression of miRNA-221. The group with miRNA-221 NC treatment served as a control. As shown in Figure 8A, the red fluorescent signal was markedly increased and the green fluorescent signal was not observed in cells transfected with miRNA-221 mimics compared to the control. However, the green fluorescence increased and the red fluorescence decreased in cells transfected with miRNA-221 inhibitors. The analyses of RT-PCR and WB further confirmed the CLSM results (Figure 8B,C), which demonstrate that the decreased expression of miRNA-221 could promote the apoptosis of Hela cells and the fluorescent nanosensor could successfully monitor miRNA regulation of apoptosis in living cells. A549 cells behaved similarly in CLSM, WB and RT-PCR assays (Figure 8D–F). These results indicate that miRNA-221 could be a promising therapeutic target for A549 and Hela cells. 

## 4. Conclusions

In summary, we designed a fluorescent nanosensor that can visualize miRNA regulation of apoptosis by simultaneously monitoring miRNA and caspase-3 in living cells. The nanosensor can circumvent the limitations of traditional methods and simultaneously detect the expressions of intracellular miRNA-221 and caspase-3 with good biocompatibility and high specificity. Confocal imaging results demonstrate that the nanosensor could be successfully used to visually evaluate the effect of miRNA-221 expression change on cell apoptosis by dual-color imaging. The prepared fluorescent nanosensor can offer a new approach for in situ monitoring of intracellular miRNA-regulated apoptosis, which is beneficial for evaluating the feasibility of miRNA-targeted therapies. We expect that the nanosensor will be a promising sensing platform for evaluating miRNA-targeting drugs and personalized therapy. 

## Data Availability

The authors confirm the data supporting the findings of this study.

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
