# Peer review of "Simultaneous Visualization of MiRNA-221 and Caspase-3 in Cancer Cells for Investigating the Feasibility of MiRNA-Targeted Therapy with a Dual-Color Fluorescent Nanosensor"

_biosensors, 2022, doi:10.3390/bios12070444_

Round 1

Reviewer 1 Report

In this manuscript, the authors developed a dual-color fluorescent nanosensor for investigating the feasibility of miRNA-targeted therapy by simultaneous visualization of miRNA-221 and caspase-3 in cancer cells. The developed nanosensor not only provides a new approach for real-time in situ monitoring the effect of miRNA-221 expression changes on caspase-3 in living cells, but also offers a promising sensing platform for evaluating miRNA targeting drugs and personalized therapy. Therefore, I recommend it publish in this journal after minor revisions.

1.    The excitation wavelength of the fluorescent dye should be added in the Figure 3 caption.

2.    Some errors should be corrected. In page 2, line 71, “Simultaneously” should be “Simultaneous”; in page 3, line 82, “1h” should be “1 h”; in page 4, line 139, “were” should be “was”.

3.    miRNA-221” in Figure 4 and Figure 5 should be “target-221”.

Reviewer 2 Report

The authors designed a fluorescent nanosensor that could visualize miR250 NA-regulation of apoptosis by monitoring miRNA and caspase-3 in live cells. Dual color confocal imaging results demonstrated the applicability of the developed system. Personally, I think there is a need for minor revision for publication, the manuscript contains valuable thoughts, I have read a well-written manuscript.

Based on the importance of the topic, I think more than 33 references can be included. My only (minor) comment is to ask the authors to add more relevant references to the manuscript.

Reviewer 3 Report

The authors designed a fluorescent nanosensor which can visualize miRNA regulation of apoptosis by simultaneously monitoring miRNA and caspase-3 in living cells. This work has a certain novelty, but a lot of experiments are needed in the introduction and material characterization. In general, this manuscript is well prepared with novelty, however, some possible confusions should be clarified before publish. Therefore, I suggest minor revision before being accepted by biosensors.

1.      The nanosensor has only been characterized by simple morphology, and some structural characterizations can be supplemented. Such as XRD, Raman, X-ray photoelectron spectroscopy (XPS). Optical performance test: excitation dependence, emission spectrum and pH tolerance. Related characterization can refer to the literature: Nanoscale Horiz. 2020, 5, 928–933. Science Advances, 2020, 6, eabb6772.

2.      The introductory part is too simplistic and does not introduce the current fluorescent probe materials.

3.      Scheme 1 still needs to be well designed, and currently it does not show the main idea of this article.

4.      Figure 2 lacks the ruler.

5.      There are some grammatical problems in the writing of the thesis, and it is recommended to revise it carefully.

Round 2

Reviewer 3 Report

There is basically no problem with the revised manuscript, and there are still some small details that need to be corrected slightly.

1.The ruler of the AFM map (Figure 2) is too ugly, it is recommended to modify it.
